# RADAR-GUIDED POLYNOMIAL FITTING FOR METRIC DEPTH ESTIMATION

## ABSTRACT

We propose POLAR, a novel radar-guided depth estimation method that introduces polynomial fitting to efficiently transform scaleless depth predictions from pre-trained monocular depth estimation (MDE) models into metric depth maps. Unlike existing approaches that rely on complex architectures or expensive sensors, our method is grounded in a fundamental insight: although MDE models often infer reasonable local depth structure within each object or local region, they may misalign these regions relative to one another, making a linear scale and shift (affine) transformation insufficient given three or more of these regions. To address this limitation, we use polynomial coefficients predicted from cheap, ubiquitous radar data to adaptively adjust depth predictions non-uniformly across depth ranges. In this way, POLAR generalizes beyond affine transformations and is able to correct such misalignments by introducing inflection points. Importantly, our polynomial fitting framework preserves structural consistency through a novel training objective that enforces local monotonicity via first-derivative regularization. POLAR achieves state-of-the-art performance across three datasets, outperforming existing methods by an average of 24.9% in MAE and 33.2% in RMSE, while also achieving state-of-the-art efficiency in terms of latency and computational cost.

## 1 INTRODUCTION

Metric 3D reconstruction is critical for spatial tasks such as self-driving (Maier et al., 2012; Gupta et al., 2021), where it is necessary for one to perceive the structure of the 3D environment in order to navigate. In many such systems, multiple sensors—including cameras, lidar, and radar—provide complementary information about the 3D scene. While lidar sensors offer dense and precise point clouds, they are expensive and not widely available (Raj et al., 2020). In contrast, radar sensors, particularly millimeter wave (mmWave) radars (Iizuka et al., 2003), return only about a hundred points per frame and are noisier (Han et al., 2023). Yet, they offer key advantages: they are far more cost-effective and energy-efficient, robust to challenging environmental conditions, and ubiquitously equipped on modern vehicles (Eichelberger & McCartt, 2016).

Unless camera baselines are known, images offer scaleless reconstruction. However, they are high-dimensional and sensitive to variations in illumination, object appearance, orientation, and camera viewpoint. Training robust models that can generalize across these factors often demands large-scale datasets, which are costly to collect. As a result, leveraging pretrained monocular depth estimation (MDE) foundation models (Ranftl et al., 2021; Yang et al., 2024; Bochkovskii et al., 2024; Piccinelli et al., 2025) emerges as a practical alternative. However, as monocular 3D reconstruction is inherently ill-posed, they typically infer scaleless relative depth, or depth that lacks the fidelity needed for applications demanding accurate metric-scale reconstruction, such as mapping and navigation.

Existing approaches (Yin et al., 2023; Viola et al., 2024; Hu et al., 2024; Zeng et al., 2025; Ding et al., 2025; Yu et al., 2025) attempt to transform these MDE predictions into accurate metric depth maps using a global affine (scale-and-shift) transformation. This class of methods assumes that the reconstruction is off by a single scaling factor across the entire scene. While effective in estimating ordinal relationships within local regions, where MDE predictions typically exhibit reasonable local (object-level) reconstructions, such linear corrections fail when multiple objects are placed at incorrect depths relative to each other. Specifically, once an MDE model places three or more objects at incorrect relative depths, no simple scale-and-shift can reconcile this misalignment (Fig. 1).

In this work, we challenge the common assumption that scale ambiguity in MDE is only up to an unknown global scale and shift. Instead, we propose to incorporate higher-order corrections through a polynomial transformation that allows for stretching and compressing at different depth levels, enabling more flexible non-uniform adjustments. Our approach, POLAR, transforms the predictions of pretrained MDE models by (i) learning prototypical patterns in the configuration of radar points, (ii) establishing spatial correspondences between radar and MDE features to encode a unified multi-sensor scene representation, and (iii) predicting polynomial coefficients that adaptively fit scaleless depth into metric depth. This polynomial expansion introduces additional degrees of freedom

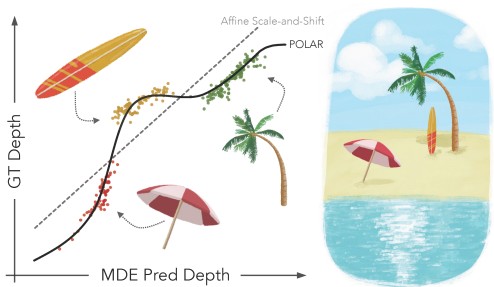

Figure 1: If an MDE model predicts incorrect relative depths between *three or more objects*, an affine scale-and-shift (dashed) cannot resolve this misalignment. POLAR (solid) overcomes this limitation by learning an $N$'th-order polynomial fit with up to $N - 2$ inflection points.

(i.e., multiple inflection points) that can better correct non-uniform variations and cross-region misalignments. While one may adjust every prediction (pixel) of an MDE model (Li et al., 2024a), the number of degrees of freedom is the total number of pixels. Given radar point clouds of a few hundred points (many orders less than the number of pixels), this becomes an ill-posed problem. On the other hand, the number of degrees of freedom in polynomial expansion is limited to the number of polynomial terms, which we empirically found to be close to the cardinality of radar point clouds, allowing our solution to be better posed and more regular. To our knowledge, no prior work has explored polynomial fitting for adapting predictions of pretrained foundation models.

However, learning such a flexible transformation is challenging because higher-order polynomials introduce a vast function space with many free parameters. Unlike a simple linear fit that relies on only scale and bias terms, polynomial fitting allows for numerous inflection points and nonlinearities. This expressiveness can inadvertently lead to harmful non-monotonic transformations—where the relative depth orderings of points are incorrectly reversed—if not properly constrained. We address this by introducing a regularization term that enforces the predicted metric depth with respect to the input MDE depth to remain approximately monotonically increasing. This ensures that incremental changes in the input depth result in proportional changes in the predicted depth within local regions where MDE reconstructs structure up to a relative scale. In essence, this regularizer encourages a *piecewise* monotonic transformation, mitigating unstable oscillations that can arise from overfitting high-degree polynomials while still enabling necessary corrections for cross-region misalignments.

**Our contributions**: We propose (1) POLAR, a novel POLynomial fitting method that leverages complementary radAR guidance to transform scaleless monocular depth into accurate metric depth. Our approach (2) introduces a fundamental insight: using polynomial coefficients predicted from a learned multimodal representation to enable non-uniform corrections. We present (3) a principled geometric formulation, where polynomial transformations introduce inflection points that can correct misalignments between local areas that an affine transformation cannot. We design (4) a novel training objective that encourages monotonicity through first-derivative as regularization, preserving local ordinality while allowing cross-region adjustments. Finally, (5) extensive experiments demonstrate that POLAR achieves the state of the art in both performance and efficiency, outperforming existing methods by an average of 29.1% and simultaneously delivering real-time processing of over 40 fps.

## 2 RELATED WORK

**Monocular Depth Estimation.** With recent advances in the scalability of neural networks (Dosovitskiy et al., 2021; Caron et al., 2021), monocular depth estimation (MDE) models can infer scaleless relative depth from a single image across diverse, unseen domains (Ranftl et al., 2020; Yang et al., 2024; Piccinelli et al., 2025; Bochkovskii et al., 2024), benefiting from large-scale datasets. Recent works seek to enhance MDE by leveraging auxiliary image signals, such as structure and motion priors from segmentation (Hoyer et al., 2023; Bian et al., 2019), uncertainty estimation (Poggi et al., 2020), optical flow (Zhao et al., 2020), and visual odometry (Song et al., 2023), to improve relative depth estimation. However, monocular *metric* depth estimation inherently suffers from scale

ambiguity, as estimating absolute depth from a single image is an ill-posed problem. Even MDE models trained with metric depth supervision often struggle to generalize to unseen domains with high fidelity (Viola et al., 2024). One way to address this limitation is to incorporate range-sensing modalities such as lidar (Jaritz et al., 2018; Ezhov et al., 2024), radar (Singh et al., 2023), or visual-inertial odometry (Wong et al., 2020).

**Image-Guided Depth Completion.** Most often studied in the context of lidar-camera depth estimation, image-guided depth completion leverages the strengths of each modality: images offer dense visual context and structural priors, while lidar points provide absolute metric scale to resolve depth ambiguity. Fusion strategies for these modalities include early fusion, where feature maps are concatenated at initial layers (Ma & Karaman, 2018; Ma et al., 2018), late fusion, where inputs are processed by independent branches (Yan et al., 2021; Rim et al., 2025), and multi-scale fusion, which captures both local details and global scene structure (Li et al., 2020). U-Net-like architectures have been widely used for coarse-to-fine depth completion (Hu et al., 2021; Lin et al., 2022), with improvements from deformable convolutions (Park et al., 2020; Xu et al., 2020), and attention mechanisms (Rho et al., 2022; Zhang et al., 2023).

**Radar-Camera Depth Estimation.** While lidar-based depth estimation methods achieve high accuracy due to their dense and precise measurements, their widespread adoption is limited by high costs, power consumption, and sensitivity to environmental conditions (Raj et al., 2020). In contrast, radar provides a cost-effective alternative (Hunt et al., 2024), offering robustness in adverse conditions such as low light, fog, and rain—where both cameras and lidar often struggle (Paek et al., 2022; Srivastav & Mandal, 2023). The ubiquity of radar sensors (Eichelberger & McCartt, 2016) in existing automotive and robotic platforms further supports their integration into depth estimation pipelines. Leveraging radar for metric depth estimation not only reduces costs but also enhances the robustness and scalability of perception systems, making it an attractive choice for real-world deployment.

Despite these advantages, methods that fuse image and radar inputs for depth estimation must address the sparsity and elevation ambiguity (Singh et al., 2023) of radar point clouds. (Lin et al., 2020) uses a two-stage late fusion approach that first produces a coarse depth map and performs outlier rejection, then predicts the final depth map. (Long et al., 2021) leverages Doppler velocity and optical flow to associate radar points with image pixels. (Lo & Vandewalle, 2021) refines radar depth using height-extension of radar points to address elevation ambiguity. (Singh et al., 2023) introduces RadarNet, which uses radar-pixel correspondence scores as well as confidence scores to generate a semi-dense depth map, which is then used to predict the final dense depth map using gated fusion. (Li et al., 2024b) mitigates distribution artifacts using sparse supervision, while (Sun et al., 2024) employs a two-stage confidence-driven approach. GET-UP (Sun et al., 2025) uses attention-enhanced graph neural networks to capture both 2D and 3D features from radar data and leverages point cloud upsampling to refine radar features. Like our method, RadarCam-Depth (Li et al., 2024a) and TacoDepth (Wang et al., 2025) also begin with an initial MDE prediction, but refine it and decode dense depth directly from fused features, rather than employing a learned scene-fitting approach.

Existing methods operate within the paradigm of completion or direct decoding of fused features, and they often rely on multi-stage training and explicit radar-pixel association learning, increasing model complexity and computational overhead. In contrast, POLAR employs a streamlined yet powerful architecture that directly predicts polynomial coefficients from radar and MDE features, enabling accurate and efficient fitting of scaleless depth to metric scale. It bypasses the need for multi-stage systems that learn explicit correspondences while enabling flexible scene-adaptive depth corrections.

## 3 METHOD FORMULATION

We aim to reconstruct a 3D scene from an RGB image $I \in \mathbb{R}^{H \times W \times 3}$ and a synchronized radar point cloud $C \in \mathbb{R}^{N_C \times 3}$ by fitting a scaleless depth map from a pretrained MDE foundation model to a metric depth map. Our method leverages MDE to seed scaleless depth predictions and estimates metric depth through polynomial fitting guided by cross-modal features that encode spatial correspondences between the 3D radar points and the dense scaleless depth predictions.

As MDE models are trained on millions of images, they are robust to typical visual nuisance variability, spanning from illumination changes and object appearances to viewpoint shifts. To circumvent the need to collect many large-scale synchronized radar-camera datasets, we begin with a

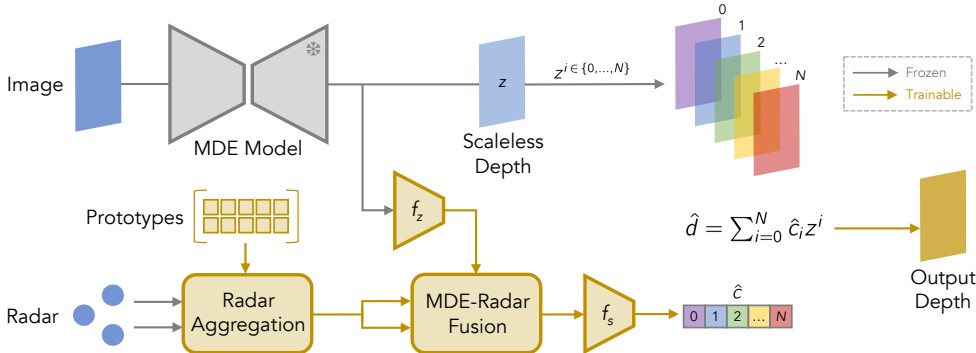

Figure 2: **Method Overview.** POLAR transforms scaleless MDE predictions into metric depth using polynomial fitting guided by radar features. Learnable prototypes extract patterns in the configurations of radar point clouds and are used to aggregate spatially-informed radar features. The geometry-aware MDE features are fused with the radar features via a learnable soft-correspondence module to yield a unified scene representation that is used to predict polynomial coefficients for fitting. This enables non-uniform corrections that improve accuracy beyond affine transformations.

frozen pretrained MDE model $M$, which serves as a learned geometric prior, to infer a scaleless depth map $z \in \mathbb{R}_+^{H \times W}$ from the image. We then encode the point cloud $C$ and the scaleless depth map $z$ separately. Through a fusion process that captures correspondences between them, we predict $N + 1$ polynomial coefficients $\{\hat{c}_0, \hat{c}_1, \cdots, \hat{c}_N\}$ that are used to transform $z$ into a metric depth map $\hat{d}$.

### 3.1 MOTIVATION

Existing methods (Zeng et al., 2025; Ding et al., 2025; Yu et al., 2025; Viola et al., 2024; Hu et al., 2024; Yin et al., 2023) refine MDE predictions by applying simple scale-and-shift transformations. However, this approach lacks the complexity needed to correct non-uniform variations from ground truth across different depths. A practical illustration of this shortcoming arises when multiple objects or local regions appear in the same scene: MDE may accurately infer the reconstruction within each object region, yet place them at incorrect relative depths with respect to each other. Such cross-region misalignments cannot be corrected by a single global scaling factor (see Fig. 1). This violates the assumption that the MDE reconstruction is up to an unknown global scale and shift.

To address this, our *polynomial fitting* approach exponentiates the MDE prediction to higher powers and learns coefficients that transform it into a metric depth map via summation. Lower-order coefficients (including scale and shift) capture the global scene layout, while higher-order coefficients focus on local depth adjustments, correcting cross-object misalignments and fine-grained MDE errors.

An intuitive understanding of polynomial fitting can be drawn from the geometric perspective of depth transformations. An affine (scale-and-shift) operation, mathematically represented as $\hat{a}z + \hat{b}$, uniformly stretches or compresses the entire reconstruction by the same factor across all depth levels. While effective for coarse global corrections, such an affine transformation has zero *inflection points*, limiting its flexibility to address cross-region misalignments in relative depth predictions. In contrast, the total degree of our polynomial fitting method determines the maximum number of potential inflection points, where the curvature of the depth transformation can change. For a polynomial $f(z) = \sum_{i=0}^{N} \hat{c}_i z^i$, an *inflection point* $z^*$ occurs if

$$f''(z^*) = \sum_{i=2}^{N} i(i-1) \hat{c}_i (z^*)^{i-2} = 0, \tag{1}$$

indicating where the second derivative changes sign. By learning inflection points, we can model transition regions in the curvature of the MDE error, allowing necessary non-uniform corrections.

### 3.2 REPRESENTATION LEARNING AND FUSION

**Radar Processing.** We begin with a radar point cloud $C \in \mathbb{R}^{N_C \times 3}$, where each of the $N_C$ points is represented by its $(x, y, z)$ coordinates in three-dimensional camera space. Radar point clouds, while

noisy and sparse, provide metric depth measurements. We concatenate a sinusoidal 3D positional embedding $\phi_{3D}(x, y, z)$ to $C$, and feed the resulting representation into a multilayer perceptron (MLP) $\psi_r$, producing radar features $F_r \in \mathbb{R}^{N_C \times c_r}$, where $c_r$ is the feature dimension.

**Radar Aggregation.** To facilitate effective radar feature aggregation, we introduce a set of learnable prototypes $P \in \mathbb{R}^{N_P \times c_r}$, where these $N_P$ prototypes capture diverse spatial and geometric properties present in the radar point cloud. Each prototype learns to focus on specific patterns in the configuration of radar points, making them highly expressive and adaptable across varying scenes. Unlike existing works (Singh et al., 2023; Li et al., 2024a) that directly encode and pool radar points—making them more susceptible to nuisances like multipath propagation—our learnable prototypes identify and match meaningful patterns within the point cloud. This allows for selective feature aggregation that mitigates the impact of outliers, resulting in more robust depth predictions.

Next, learnable prototypes are matched to the most relevant radar features for a given input. We treat the prototypes as centroids and perform soft clustering over the radar features. Each radar point is softly assigned to prototypes according to feature similarity, and the corresponding values are aggregated to yield a global scene-descriptive representation $F_R$:

$$D_{ij} = \|P_j - \Phi_r(F_r)_i\|^2, \quad F_R = \text{softmax}(-D / \tau) \ \Psi_r(F_r), \tag{2}$$

where $\Phi_r, \Psi_r$ are MLPs that project the radar features $F_r$, the matrix $D \in \mathbb{R}^{N_P \times N_C}$ contains the pairwise squared distances between prototypes $\{P_j\}_{j=1}^{N_P}$ and projected radar features $\{\Phi_r(F_r)_i\}_{i=1}^{N_C}$, and $\tau$ denotes temperature. This clustering-based formulation enables prototypes to capture recurring spatial and geometric patterns in radar point clouds. Comparatively, applying lidar depth completion methods, which densify a sparse projection of points using surrounding context, to a radar point cloud results in poor performance (see Sec. E). This is because radar measurements are orders of magnitude sparser than lidar and are much noisier, especially when lacking sufficient antenna elements, e.g., elevation ambiguity, or range resolution (Singh et al., 2023).

**MDE-Radar Fusion.** The scaleless depth map $z \in \mathbb{R}_+^{H \times W}$ from the MDE model is encoded with a learnable depth encoder $f_z$, producing depth features $Z \in \mathbb{R}^{(H \times W) \times c_z}$. These features inherit invariants learned through large-scale MDE training—such as robustness to color variations, illumination changes, and diverse object poses—and therefore primarily encode object-level properties. Since variations in the shape of objects and their geometry are generally more stable across scenes than photometric appearance, the resulting depth features provide a more reliable geometric context for fusion with radar features. This allows radar point configurations to be matched to consistently observed shapes and surfaces, rather than to pixel-wise color intensities that can vary arbitrarily with lighting and viewpoint. To this end, we learn soft spatial correspondences between the depth features $Z$ and the radar features $F_R$ to construct a unified scene representation $S$ that fuses the structural information encoded in the MDE predictions with the metric cues provided by the radar features:

$$S = \text{softmax}\left(\frac{(Z + E) \times (\Phi_R(F_R))^T}{\sqrt{c_r}}\right) \Psi_R(F_R), \tag{3}$$

where $E \in \mathbb{R}^{(H \times W) \times c_z}$ is a learned 2D positional embedding, and $\Phi_R, \Psi_R$ are MLPs that project the aggregated radar features $F_R$. $c_z$ is set equal to $c_r$ to align the depth and radar features within a common embedding space, facilitating cross-modal fusion, and $S$ is reshaped to be in $\mathbb{R}^{H \times W \times c_z}$.

**Predicting Coefficients.** The fused representation $S$ is passed through a shallow convolutional neural network (CNN) $f_s$ followed by a global average pooling (GAP) layer to yield $\bar{S} = \text{GAP}(f_s(S))$, which is a $c_s$-dimensional feature vector. The GAP layer aggregates spatial information from the entire scene, ensuring that $\bar{S}$ captures global context. This final scene representation $\bar{S}$ is then fed into an MLP $\psi_s$, which predicts the $N + 1$ polynomial coefficients as a vector $\hat{c}$:

$$\hat{c} = \psi_s(\bar{S}) \in \mathbb{R}^{N+1}. \tag{4}$$

These coefficients allow the model to adaptively refine the initial depth predictions from the MDE model, with each coefficient adjusting depth at different scales and granularities. Lower-order terms capture broad scene structure, while higher-order terms enable fine-grained and cross-region corrections, resulting in a high-fidelity metric-scale depth map.

## 3.3 POLYNOMIAL FITTING

Given a scaleless depth map $z \in \mathbb{R}_+^{H \times W}$ predicted by the MDE foundation model, our goal is to transform it into a metric depth map $\hat{d} \in \mathbb{R}_+^{H \times W}$ using the polynomial coefficients $\{\hat{c}_0, \hat{c}_1, \ldots, \hat{c}_N\}$ predicted by our network (Sec. 3.2). Formally, we express the final depth map $\hat{d}$ as:

$$\hat{d}(x,y) = \sum_{i=0}^{N} \hat{c}_i \cdot z(x,y)^i, \quad \forall (x,y) \in H \times W. \tag{5}$$

This polynomial formulation applies the learned coefficients to successive powers of the scaleless depth map, enabling complex, non-linear transformations that surpass the limitations of affine transformations. Each pixel's final depth is thus computed by summing multiple weighted terms, where each term corresponds to a different polynomial order of the initial depth prediction.

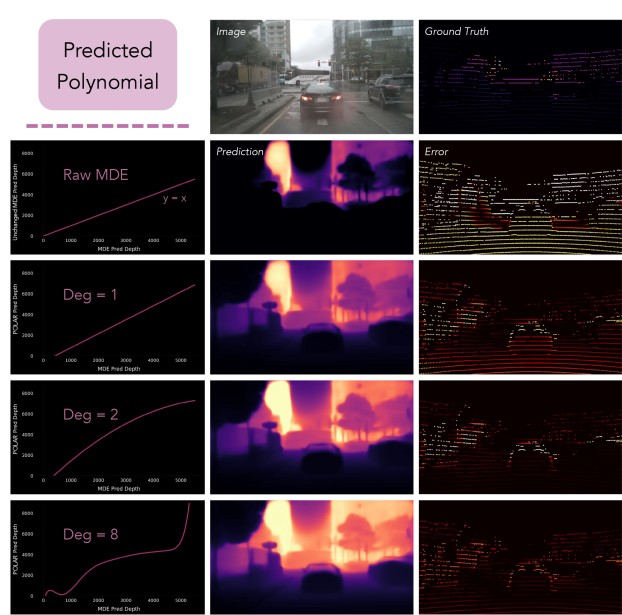

This is in contrast to directly mapping to a metric-scale depth map (Singh et al., 2023; Li et al., 2024a), where the degrees of freedom comprise every predicted pixel. Each corresponding point and pixel prediction can be viewed as a constraint or equation in our estimation system. As the radar point cloud ($\sim 10^2$ points) only occupies a small subset of the image space ($\sim 10^6$ pixels), the solution is underdetermined and often irregular (see Fig. 4). Conversely, a simple linear fit (only two degrees of freedom) with inconsistent MDE predictions and noisy radar points leads to an

Figure 3: POLAR leverages spatial information from radar points to predict higher-degree polynomial transformations that can correct non-affine errors in MDE predictions.

overdetermined system (see Tab. 5). POLAR offers the middle ground: By a flexible choice of polynomial degree, we can appropriately tune and select the complexity of the function, i.e., degrees of freedom, that leads to a better fit between MDE predictions and radar points.

In terms of overhead, compared to predicting two coefficients (scale and shift) for an affine transformation, predicting $N + 1$ coefficients for polynomial fitting introduces a negligible increase in computational cost ($< 0.01\%$ more FLOPs): the final MLP outputs an $(N + 1)$-dimensional vector instead of a two-dimensional vector, and the final depth map is computed through parallelized exponentiation and multiplication operations (see Sec. B for the complete derivation).

**Geometric Intuition.** Applying polynomial fitting to the scaleless depth map is best viewed as a flexible, depth-dependent correction mechanism. In contrast to a single scale-and-shift operation, which uniformly adjusts the entire depth field by a single global factor, polynomial transformations introduce multiple degrees of freedom—higher-order terms enable correction of misalignments in relative depth between objects and local regions.

By introducing polynomial terms, our method enables depth adjustments that vary according to the initial depth estimates $z$. Lower-order coefficients ($\hat{c}_0, \hat{c}_1, \cdots$) establish a broad global scale, providing a metric basis for the initially scaleless $z$. Higher-order coefficients ($\cdots, \hat{c}_{N-1}, \hat{c}_N$) introduce both low-frequency corrections, e.g., resolving cross-object misalignments (Fig. 1), as well as high-frequency corrections, e.g., sharpening fine object boundaries (Fig. 4, Image C).

The concept of inflection points provides an intuitive understanding of this approach. Each additional polynomial order introduces more potential inflection points, allowing the depth transformation to shift curvature where needed. This flexibility enables "stretching" or "compressing" different depth levels: areas already near their correct metric depth receive minimal adjustment, while regions

Table 1: **Quantitative results** on the nuScenes, ZJU-4DRadarCam (ZJU), and View-of-Delft test sets evaluated with various maximum evaluation distances.

| Distance | Method | | nuScenes | | ZJU | | View-of-Delft | |
|---|---|---|---|---|---|---|---|---|
| | | | MAE | RMSE | MAE | RMSE | MAE | RMSE |
| 50m | RadarNet | [CVPR '23] | 1727.7 | 3746.8 | 1430.5 | 3250.8 | 2944.8 | 6113.2 |
| | SparseBeatsDense | [ECCV '24] | 1524.5 | 3567.3 | 1424.4 | 3267.5 | 2909.9 | 5746.4 |
| | GET-UP | [WACV '25] | 1241.0 | 2857.0 | 1483.9 | 3220.5 | 2330.0 | 4565.0 |
| | RadarCam-Depth | [ICRA '24] | 1286.1 | 2964.3 | 1067.5 | 2817.4 | 1895.1 | 4458.7 |
| | TacoDepth | [CVPR '25] | 1046.8 | 2487.5 | 930.2 | 2477.3 | - | - |
| | **POLAR (Ours)** | | **1014.4** | **2475.7** | **578.0** | **1108.6** | **1293.7** | **2420.6** |
| 70m | RadarNet | [CVPR '23] | 2073.2 | 4590.7 | 1543.8 | 3655.3 | 3428.9 | 7331.9 |
| | SparseBeatsDense | [ECCV '24] | 1822.9 | 4303.6 | 1520.0 | 3593.4 | 3408.3 | 6914.7 |
| | GET-UP | [WACV '25] | 1541.0 | 3657.0 | 1651.5 | 3711.7 | 2758.0 | 5678.0 |
| | RadarCam-Depth | [ICRA '24] | 1587.9 | 3662.5 | 1157.0 | 3117.7 | 2095.4 | 4944.6 |
| | TacoDepth | [CVPR '25] | 1347.1 | 3152.8 | 983.1 | 2779.6 | - | - |
| | **POLAR (Ours)** | | **1286.1** | **2947.3** | **603.7** | **1154.9** | **1442.9** | **2803.8** |
| 80m | RadarNet | [CVPR '23] | 2179.3 | 4898.7 | 1578.4 | 3804.2 | 3597.0 | 7809.2 |
| | SparseBeatsDense | [ECCV '24] | 1927.0 | 4609.6 | 1548.4 | 3708.1 | 3584.0 | 7375.2 |
| | GET-UP | [WACV '25] | 1632.0 | 3932.0 | 1699.7 | 3882.6 | 2917.3 | 6145.1 |
| | RadarCam-Depth | [ICRA '24] | 1689.7 | 3948.0 | 1183.5 | 3229.0 | 2227.4 | 5385.8 |
| | TacoDepth | [CVPR '25] | 1492.4 | 3324.8 | 1032.5 | 2850.3 | - | - |
| | **POLAR (Ours)** | | **1407.8** | **3193.5** | **629.6** | **1171.3** | **1500.1** | **3951.8** |

suffering larger errors (e.g., due to misalignment with other regions) undergo more substantial correction. Higher-order terms magnify small (i.e., high-frequency) discrepancies in the initial MDE predictions, making them more apparent to the model and easier to correct. In this way, the degree of the polynomial, selected as a hyperparameter, governs the capacity of our model to apply non-uniform corrections across the scene.

The sign of the predicted polynomial coefficients furthers our interpretation. Positive coefficients for higher-order terms push depth values outward while negative ones pull them inward, effectively expanding or contracting selected depth intervals to correct local errors. In doing so, these coefficients shape the curvature of the polynomial, dictating where and how inflection points arise. For instance, if the model consistently overestimates distant objects, a negative high-order coefficient can compress that region, whereas a positive coefficient might be learned to correct underestimations. By treating coefficient signs as dynamic anchors for curvature changes, the polynomial fitting framework provides an interpretable mechanism (e.g., Fig. 3) for refining scaleless depth into high-fidelity metric depth.

### 3.4 LOSS FUNCTION

We employ a loss function comprising three terms, each weighted by its $\lambda$, to guide the learning of the polynomial coefficients and ensure accurate depth estimation. Our loss is defined as:

$$\mathcal{L} = \lambda_{L_1}\|\hat{d} - d_{\text{gt}}\|_1 + \lambda_{L_2}\|\hat{d} - d_{\text{gt}}\|_2^2 + \lambda_{\text{mono}}\left\|\mathbf{1}_{H \times W} - \frac{\mathrm{d}\hat{d}}{\mathrm{d}z}\right\|_1, \tag{6}$$

where $d_{\text{gt}} \in \mathbb{R}_+^{H \times W}$ is the ground truth metric depth map, $\hat{d} \in \mathbb{R}_+^{H \times W}$ is our predicted metric depth, and $z \in \mathbb{R}_+^{H \times W}$ is the scaleless depth map predicted by the MDE model.

The first two terms, $L_1$ and $L_2$ losses, ensure that our predicted depth map $\hat{d}$ closely matches the ground truth $d_{\text{gt}}$ by penalizing discrepancies. The $L_1$ term is less sensitive to outliers and thus promotes robustness, while the $L_2$ term penalizes larger errors more significantly.

The novel component of our objective lies in the third term, which constrains the first derivative of the predicted depth $\hat{d}$ w.r.t the input scaleless depth $z$ (Eq. 7) to remain near that of $\hat{d} = z$.

$$\frac{\mathrm{d}\hat{d}}{\mathrm{d}z}(x,y) = \sum_{i=1}^{N} i\,\hat{c}_i\,z(x,y)^{i-1}. \tag{7}$$

This regularization enforces that our polynomial fitting function remains approximately monotonically increasing, akin to isotonic regression (Barlow et al., 1972). Generally, within an object or local

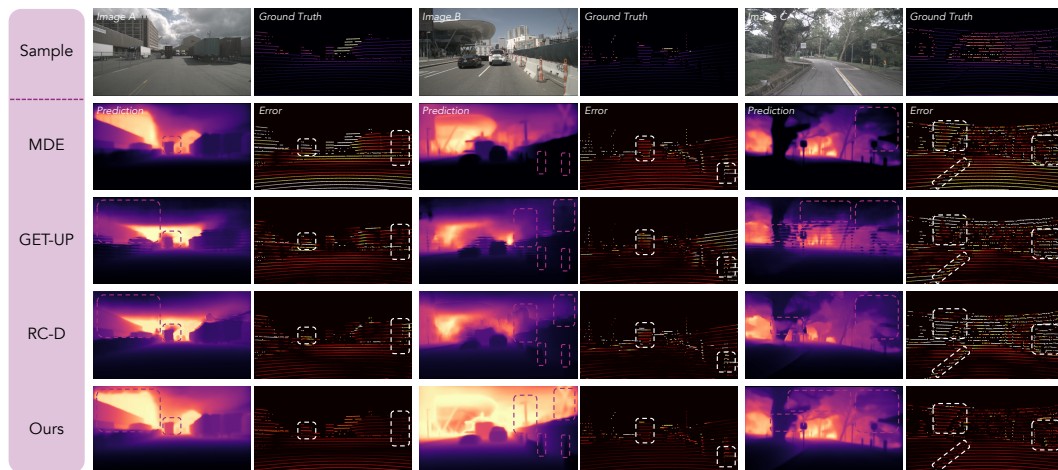

Figure 4: **Qualitative results** on nuScenes. GET-UP and RadarCam-Depth (RC-D) fail to reconstruct entire regions, yielding objects with large depth errors. Raw MDE yields reasonable relative reconstructions but suffers from incorrect global scale and cross-object misalignments. POLAR leverages polynomial fitting to recover a global scale and correct these misalignments. See Fig. 5 for colorbars.

region, a pixel with a higher initial scaleless depth value should not be assigned a lower final metric depth value compared to a pixel with a lower initial scaleless depth. The preservation of ordinal relationships is an inductive bias grounded in the assumption that the MDE model provides a reasonably accurate estimation of intra-object relative depth.

Polynomial fitting introduces a large function space, with expressiveness growing with degree $N$, which can significantly and detrimentally disrupt local monotonic depth ordering if unconstrained. Our regularization term addresses this by preserving local depth ordinality while still allowing corrections of cross-object misalignments. This constraint prevents overfitting of nonlinear transformations to noisy radar data, spurious correlations, and outliers, thereby avoiding the oscillatory behavior that is a known challenge with fitting higher-degree polynomials (Bishop, 2006).

## 4 EXPERIMENTS

**Baseline Methods.** We consider five recent radar-camera depth estimation baselines: RadarNet (Singh et al., 2023), SparseBeatsDense (Li et al., 2024b), GET-UP (Sun et al., 2025), RadarCam-Depth (Li et al., 2024a), and TacoDepth (Wang et al., 2025). Notably, both RadarCam-Depth and TacoDepth also take scaleless MDE predictions as input, together with radar points, to predict metric depth.

**Datasets.** We evaluate all methods on the nuScenes, ZJU-4DRadarCam (ZJU), and View-of-Delft (VoD) datasets (see Sec. C for more details) using the MAE and RMSE metrics with maximum evaluation distances of 50, 70, and 80 meters, following established conventions in the literature.

### 4.1 MAIN RESULTS

Compared to baselines, POLAR reduces MAE by 4.4% and RMSE by 3.7% on nuScenes, 38.5% and 57.5% on ZJU, and 31.8% and 38.5% on VoD, achieving SOTA results for all datasets (Tab. 1).

We qualitatively compare POLAR against baseline methods on the challenging nuScenes dataset, providing a visual demonstration of its improved depth estimation performance (Fig. 4). POLAR more accurately predicts metric scale and exhibits fewer global misalignments. Furthermore, it is able to extrapolate depth predictions of object surfaces from learned photometric priors, an ability inherited from the backbone MDE model (e.g., UniDepth) trained on large-scale datasets. Our polynomial fitting approach then non-uniformly corrects depth discrepancies across the scene, leading to significant performance improvement over initial MDE predictions. Introducing multiple degrees of freedom enables POLAR to correct misalignments of objects relative to each other, improving accuracy by correcting consistent over- or under-estimations in local regions. For instance, the raw MDE prediction for Image C incorrectly places the bus stop roof at around the same depth as the

more distant tree branches, as highlighted on the right. Additionally, MDE does not accurately infer the boundary between the curb and asphalt, again placing both at erroneously similar depths. With a learned polynomial transformation, POLAR corrects these misalignments with depth-dependent adjustments, accurately separating the curb from the asphalt and placing the bus stop roof and tree branch at their correct depths.

While RadarCam-Depth also utilizes an MDE backbone, its complex processing of raw MDE predictions can be seen to distort initially correct depth structure. As shown in the top-left highlighted region of Image C, the tree trunk and attached fern are incorrectly placed at different depths despite the initial MDE predictions correctly positioning them at similar depths. Furthermore, overfitting to specific regions within a scene—particularly those with denser ground truth—may reduce overall error relative to the MDE model but can lead to unintended geometric artifacts. This can result in the omission of entire structures in the predicted depth map, such as the highlighted building on the left of Image A and the construction crane arms in Image B. GET-UP also struggles to accurately infer depth structure, resulting in the omission of objects across all three samples: the highlighted building in Image A, the crane arms in Image B, and foreground tree branches in Image C.

As the polynomial degree—selected as a hyperparameter—increases, the maximum possible number of inflection points also grows, giving POLAR greater expressive power to correct misalignments between the many scene elements in Fig. 3, including cars, trucks, trees, buildings, and traffic lights. This is reflected in the successively improved error maps. See Sec. A for quantitative evaluations.

## 4.2 Computational Efficiency

Table 2: Training times (minutes per epoch) on nuScenes using an NVIDIA A6000 GPU.

| Method | Train Time / Epoch |
|---|---|
| Lin (Lin et al., 2020) | 89.25 |
| RadarNet (Singh et al., 2023) | 101.50 |
| SparseBeatsDense (Li et al., 2024b) | 63.96 |
| GET-UP (Sun et al., 2025) | 249.57 |
| RadarCam-Depth (Li et al., 2024a) | 86.38 |
| **POLAR (Ours)** | **33.16** |

Table 3: Inference time in milliseconds (ms) and computational cost (GFLOPs) on nuScenes.

| Method | Inference Time | GFLOPs |
|---|---|---|
| Lin (Lin et al., 2020) | 129.96 | 550.43 |
| SparseBeatsDense (Li et al., 2024b) | 97.47 | 532.74 |
| GET-UP (Sun et al., 2025) | 445.45 | 630.99 |
| RadarCam-Depth (Li et al., 2024a) | 315.64 | 619.02 |
| TacoDepth (Wang et al., 2025) | 29.30 | 139.87 |
| **POLAR (Ours)** | **24.81** | **89.70** |

POLAR achieves state-of-the-art training time, inference time (ms), and computational overhead (GFLOPs) compared to all baseline methods, while also achieving state-of-the-art accuracy. Tab. 2 shows that POLAR has the lowest training time per epoch among all methods, requiring 33.16 minutes per epoch on nuScenes. This efficiency stems from our streamlined design, which avoids multi-stage processing and explicit radar-camera association learning, both of which contribute to longer training times in methods such as RadarCam-Depth and GET-UP.

Tab. 3 highlights POLAR's state-of-the-art inference speed, requiring just 24.81 ms per frame—a reduction of 15.3% compared to the previous state-of-the-art TacoDepth and 92.1% compared to RadarCam-Depth. This corresponds to 40.3 fps, enabling real-time depth perception. Furthermore, POLAR achieves this inference speed with a lower computational overhead of 89.70 GFLOPs—a 39.5% reduction relative to TacoDepth and an 85.5% reduction relative to RadarCam-Depth. Taken together, these results establish POLAR as not only the most accurate radar-camera depth estimation method, but also the most practical for real-time deployment where latency is critical.

## 5 Discussion

**Limitations.** POLAR requires tuning the polynomial degree as a hyperparameter (Sec. A), which may vary across datasets. Additionally, while our novel first-derivative regularization mitigates oscillatory behavior from high-degree polynomials, further constraints could enhance stability.

**Summary.** We are the first to formulate radar-camera depth estimation as a scene-fitting problem, leveraging high-degree polynomials to transform MDE predictions into accurate metric depth. Our principled and efficient approach demonstrates that a fundamental insight—shifting from affine scale-and-shift to flexible polynomial transformations—outperforms computationally heavier methods.

ETHICS STATEMENT

As our work focuses on depth estimation, we do not anticipate any direct ethical concerns regarding the proposed method. However, as with any data-driven approach, the model may be biased towards performing well on data distributions similar to those seen during training and may underperform in out-of-distribution or underrepresented scenarios.

REPRODUCIBILITY STATEMENT

Our methodology is fully described in Sec. 3, and the evaluation metrics and implementation details we use are provided in Sec. F. We plan to release code for full reproducibility. In Sec. H, we provide a proof by construction that demonstrates the limitations of global scale and shift alignment in monocular depth estimation.

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

APPENDIX

## A COMPARATIVE STUDIES

**MDE Models.** We evaluate POLAR using four monocular depth estimation (MDE) backbones: DPT (Ranftl et al., 2021), Depth Anything (Yang et al., 2024), Depth Pro (Bochkovskii et al., 2024), and UniDepth (Piccinelli et al., 2025). DPT and Depth Anything infer scaleless inverse depth, and while Depth Pro and UniDepth output metric depth, their raw predictions exhibit significant deviation from metric ground truth (see raw MDE performance in Tabs. 4, 9, 10), and thus we process them the same way as scaleless depth.

Tab. 4 compares raw MDE performance to their performance when used as backbones for RadarCam-Depth and POLAR. Raw MDE predictions show large errors due to scale ambiguity, while RadarCam-Depth provides moderate im-

Table 4: **MDE backbone** comparative studies on nuScenes.

| | nuScenes | |
|---|---|---|
| Method | MAE | RMSE |
| DPT (Ranftl et al., 2021) | 5188.2 | 6884.5 |
| Depth Anything (Yang et al., 2024) | 2404.9 | 4851.1 |
| Depth Pro (Bochkovskii et al., 2024) | 3835.0 | 6600.3 |
| UniDepth (Piccinelli et al., 2025) | 2129.8 | 4887.7 |
| RadarCam-Depth w/ DPT | 1689.7 | 3948.0 |
| RadarCam-Depth w/ Depth Anything | 1953.6 | 5107.8 |
| RadarCam-Depth w/ Depth Pro | 3417.1 | 6462.0 |
| RadarCam-Depth w/ UniDepth | 1872.0 | 4321.2 |
| POLAR w/ DPT | 1525.6 | 3745.0 |
| POLAR w/ Depth Anything | 1515.1 | 3719.4 |
| POLAR w/ Depth Pro | 1627.7 | 4143.9 |
| POLAR w/ UniDepth | **1407.8** | **3193.5** |

provements. In contrast, POLAR consistently reduces error across all backbones, demonstrating the effectiveness of polynomial fitting. For DPT and Depth Anything, outputs are first inverted when used as a backbone, and are further median-scaled for the reported raw results. Among all configurations, POLAR w/ UniDepth achieves the best performance, improving over raw UniDepth predictions by 51.2% and over RadarCam-Depth w/ UniDepth by 37.8%.

Table 5: **Sensitivity study** of polynomial degree, selected as a hyperparameter.

| | nuScenes | | ZJU | |
|---|---|---|---|---|
| Polynomial Degree | MAE | RMSE | MAE | RMSE |
| 1 (Scale + Shift) | 2156.8 | 4491.3 | 1078.2 | 2405.5 |
| 2 | 1715.2 | 3840.6 | 901.0 | 1822.7 |
| 4 | 1482.7 | 3510.1 | 791.2 | 1516.4 |
| 6 | 1466.9 | 3496.0 | 670.3 | 1297.9 |
| 8 | **1407.8** | **3193.5** | **629.6** | **1171.3** |
| 10 | 1463.7 | 3494.5 | 643.3 | 1184.5 |

Table 6: **Ablation studies** of architecture and loss components.

| | nuScenes | | VoD | |
|---|---|---|---|---|
| Ablated Component | MAE | RMSE | MAE | RMSE |
| learnable prototypes | 1615.5 | 3629.0 | 1619.3 | 4162.7 |
| cross-modality att. | 2238.8 | 4817.4 | 2147.9 | 4926.1 |
| monotonicity loss | 1921.1 | 4399.5 | 1924.5 | 4660.3 |
| pos. embeddings | 1454.1 | 3488.9 | 1500.1 | 3951.8 |
| no ablations | 1407.8 | 3193.5 | 629.6 | 1171.3 |

**Polynomial Degree.** Tab. 5 quantifies the performance gains achieved with higher-degree polynomials, with degree 8 outperforming lower degrees. At degree 10, performance degrades slightly, potentially due to excessive flexibility resulting in detrimental oscillations.

**Ablations.** We evaluate the impact of key components of POLAR in Tab. 6. Replacing learnable prototypes with self-attention on the radar point features degrades performance, demonstrating that prototypes capture relevant patterns within radar point configurations. Removing cross-modality fusion results in the largest per-

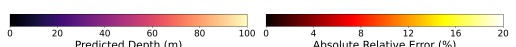

Figure 5: To quantify absolute improvement in our qualitative results, we provide the colorbars that were used in all examples in Figs. 3, 4, 6, and 7.

formance drop, highlighting the necessity of leveraging correspondences between MDE and radar features. Removing the monotonicity loss term also reduces performance, suggesting that depth ordering regularization is crucial for stable polynomial fitting.

## B MORE ON COMPUTATIONAL EFFICIENCY

**Incremental increase in computation for each additional polynomial degree.** Adding a $k+1$-th polynomial term requires minimal computational overhead. For an MDE prediction of shape $(H, W)$, the additional cost consists of three components: predicting another coefficient via the linear layer of $(1 \times 64 \times 2$ FLOPs), computing the $(k+1)$-th power from pre-computed $k$-th exponentiation

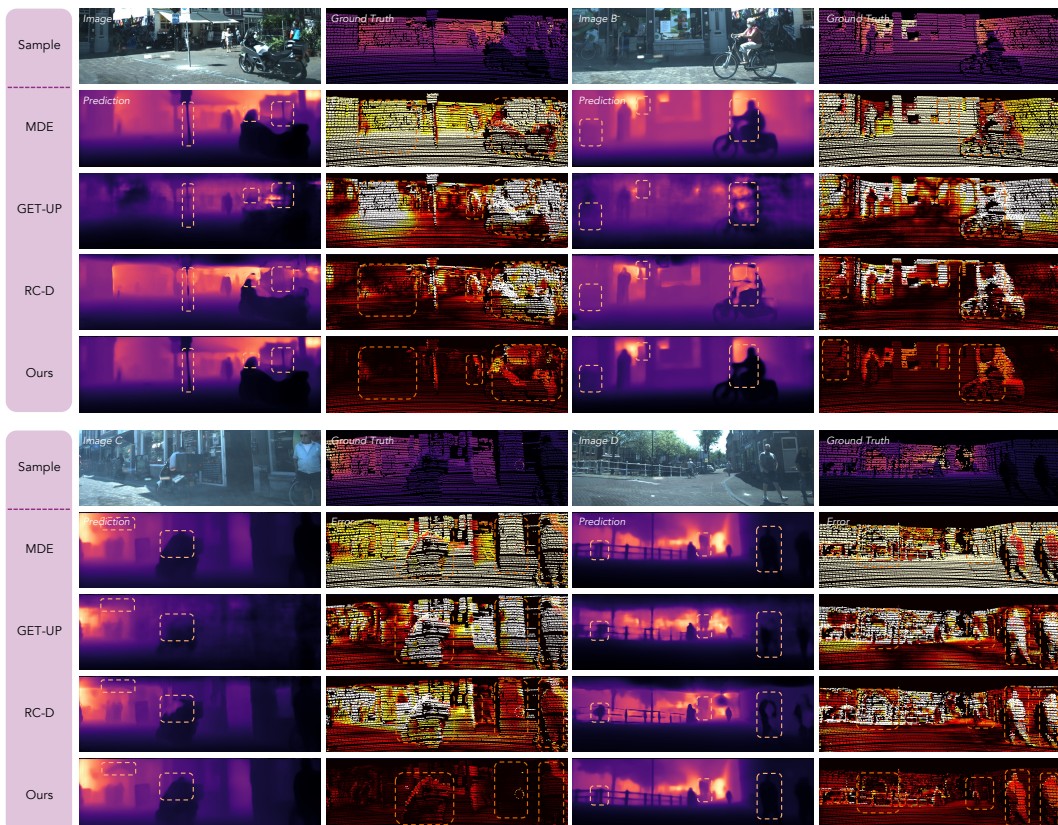

Figure 6: **Additional qualitative comparison** on VoD. POLAR more accurately predicts metric scale, and exhibits fewer global misalignments and geometric artifacts in comparison to other methods. Polynomial fitting, allowing POLAR to non-uniformly scale distinct local regions, results in notable improvements in metric depth estimation over initial MDE predictions. This is apparent in all MDE error maps: though object depth geometry is reasonable, actual metric depth values tend to be inaccurate. GET-UP, while marginally improving metric depth estimation relative to MDE, presents substantial errors in depth structure. For instance, the presence of geometric artifacts or omission of objects or regions entirely can be seen through the following highlighted examples: loss of motorcycle body and visor in Image A, front portion of biker (right) and bike wheel (far-left) in Image B, top of motorcyclist in Image C, and street sign in Image D. RadarCam-Depth (RC-D) similarly exhibits geometric artifacts, present in the sign pole, motorcycle, and fence post in Image A, biker and bicycle wheel in Image B, roof overhang and motorcyclist in Image C, and fence posts and pedestrians in Image D. In contrast, POLAR extrapolates correct structure from photometric priors while also demonstrating clear improvements in metric depth fidelity compared to all other baselines.

($H \times W$ multiplications), and incorporating this term into the final depth prediction ($2 \times H \times W$ operations for multiplication by the coefficient and addition). For nuScenes, where each image has shape $(900, 1600)$, each additional polynomial term incurs an additional computational cost of $0.0043$ GFLOPs, which is a $0.0048\%$ increase.

## C  DATASET DETAILS

The **nuScenes** dataset (Caesar et al., 2019) contains 1,000 scenes, each lasting 20 seconds, collected from a vehicle equipped with Velodyne HDL32E lidar, Continental ARS 408-21 Radar, Basler acA1600-60gc camera with $900 \times 1600$, and Advanced Navigation Spatial IMU sensors around Singapore and Boston. This data collection process resulted in 40,000 synchronized keyframes. Each frame has an average of 97 radar point measurements. Additionally, the dataset includes 877,993 3D bounding box annotations about 23 object classes, and is organized with a train-test split of 850 scenes for training and validation, and 150 for testing.

The **ZJU-4DRadarCam** (ZJU) dataset (Li et al., 2024a) provides lidar, Radar, and camera data, collected through the same method as the nuScenes dataset around Hangzhou, China. The dataset is enhanced with high-density lidar and 4D radar data, utilizing the RoboSense M1 lidar sensor and Oculii's EAGLE 4D radar sensor. Additionally, the vehicle is outfitted with RealSense D455 cameras. The dataset includes a total of 33,409 synchronized keyframes, divided into 29,312 frames for training and validation, and 4,097 frames for testing. Each frame has an average of 465 radar point measurements. The original camera resolution was $720 \times 1280$ but was cropped to $300 \times 1280$ because of the limited presence of reprojected lidar points.

The **View-of-Delft** (VoD) dataset (Palffy et al., 2022) uses similar methods to provide lidar, Radar, and camera data around the city of Delft in the Netherlands. The vehicle was equipped with a Velodyne HDL-64 S3 LIDAR, ZF FRGen 21 3+1D Radar, a stereo camera with $1216 \times 1936$ resolution, an RTK GPS, IMU, and wheel odometry. It contains 8,693 frames of synchronized and calibrated keyframes along with 123,106 3D bounding box annotations about 13 road user classes. Each frame has an average of 276 radar point measurements. Similar to the ZJU dataset, the camera resolution was cropped to $608 \times 1936$ because of the limited presence of reprojected lidar points.

## D   FULL NUSCENES BENCHMARK

We present the full set of quantitative results on the nuScenes dataset in Tab. 7. This table includes additional baseline methods that were omitted from the main text for brevity, providing a comprehensive comparison of POLAR against all known existing radar-camera depth estimation methods. As shown, POLAR consistently outperforms all competing methods across all maximum distance thresholds (50m, 70m, and 80m), achieving the lowest mean absolute error (MAE) and root mean squared error (RMSE).

## E   COMPARISON TO DEPTH COMPLETION

One potential idea for radar-camera depth estimation is to apply lidar-camera depth completion methods designed to densify sparse depth maps using surrounding context. One such method, BPNet (Tang et al., 2024) achieves state-of-the-art performance on the KITTI depth completion benchmark by leveraging bilateral propagation. However, when applied to radar-camera depth estimation, BPNet performs poorly, as shown in Tab. 7. POLAR outperforms BPNet by 57.9% in MAE and 44.2% in RMSE on nuScenes, highlighting the limitations of directly applying lidar depth completion methods to radar data. The key reason for this underperformance lies in the fundamental differences between lidar and radar point clouds. Unlike lidar, radar measurements are orders of magnitude sparser and significantly noisier due to factors such as limited antenna elements, elevation ambiguity (see Fig. 7), and lower range resolution (Singh et al., 2023). Lidar depth completion methods assume a relatively dense and structured input (Xia et al., 2023), leveraging local spatial continuity to propagate depth estimates effectively. In contrast, radar points are too sparse for such methods to infer meaningful local depth relationships, leading to poor depth reconstruction when attempting direct densification.

In addition, we compare against Non-Local Spatial Propagation Network (NLSPN) (Park et al., 2020), which achieves near state-of-the-art performance on the KITTI depth completion benchmark by refining sparse lidar depth with an iterative non-local spatial propagation procedure. NLSPN predicts an initial depth map along with pixel-wise confidences, then refines it by estimating non-local neighbors and their corresponding affinities to selectively propagate depth informa-

Table 8: **Error metrics for depth estimation.** These metrics compute the error between predicted depth $\hat{d}(x)$ and ground truth depth $d(x)$.

| Metric | Definition |
|---|---|
| MAE↓ | $\frac{1}{|\Omega|} \sum_{x \in \Omega} |\hat{d}(x) - d(x)|$ |
| RMSE↓ | $\left( \frac{1}{|\Omega|} \sum_{x \in \Omega} |\hat{d}(x) - d(x)|^2 \right)^{1/2}$ |

tion. Unlike other approaches that rely on fixed local neighbors, NLSPN adaptively determines relevant non-local neighbors, improving depth completion accuracy, especially near depth boundaries. However, again, when applied to radar-camera depth estimation, NLSPN performs poorly, as shown in Tab. 7. POLAR outperforms NLSPN by 59.8% in MAE and 55.4% in RMSE on nuScenes.

Table 7: Full quantitative results (mm) on the nuScenes benchmark.

| Distance | Method | nuScenes | |
| --- | --- | --- | --- |
| | | MAE | RMSE |
| 50m | NLSPN (Park et al., 2020) | 2790.0 | 5813.4 |
| | BPNet (Tang et al., 2024) | 2407.0 | 4438.0 |
| | RC-PDA (Long et al., 2021) | 2225.0 | 4156.5 |
| | RC-PDA-HG (Long et al., 2021) | 2210.0 | 4234.0 |
| | BTS (Lee et al., 2019) | 1937.0 | 3885.0 |
| | DORN (Lo & Vandewalle, 2021) | 1926.6 | 4124.8 |
| | RadarNet (Singh et al., 2023) | 1727.7 | 3746.8 |
| | CaFNet (Sun et al., 2024) | 1674.0 | 3674.0 |
| | Lin (Lin et al., 2020) | 1598.2 | 3790.1 |
| | SparseBeatsDense (Li et al., 2024b) | 1524.5 | 3567.3 |
| | RadarCam-Depth (Li et al., 2024a) | 1286.1 | 2964.3 |
| | GET-UP (Sun et al., 2025) | 1241.0 | 2857.0 |
| | TacoDepth (Wang et al., 2025) | 1046.8 | 2487.5 |
| | **POLAR (Ours)** | **1014.4** | **2475.7** |
| 70m | NLSPN (Park et al., 2020) | 3140.0 | 6580.6 |
| | RC-PDA (Long et al., 2021) | 3326.1 | 6700.6 |
| | RC-PDA-HG (Long et al., 2021) | 3485.6 | 7002.9 |
| | BTS (Lee et al., 2019) | 2346.0 | 4811.0 |
| | DORN (Lo & Vandewalle, 2021) | 2170.0 | 4532.0 |
| | RadarNet (Singh et al., 2023) | 2073.2 | 4590.7 |
| | CaFNet (Sun et al., 2024) | 2010.0 | 4493.0 |
| | Lin (Lin et al., 2020) | 1897.8 | 4558.7 |
| | SparseBeatsDense (Li et al., 2024b) | 1822.9 | 4303.6 |
| | RadarCam-Depth (Li et al., 2024a) | 1587.9 | 3662.5 |
| | GET-UP (Sun et al., 2025) | 1541.0 | 3657.0 |
| | TacoDepth (Wang et al., 2025) | 1347.1 | 3152.8 |
| | **POLAR (Ours)** | **1286.1** | **2947.3** |
| 80m | NLSPN (Park et al., 2020) | 3257.7 | 6872.4 |
| | RC-PDA (Long et al., 2021) | 3721.0 | 7632.0 |
| | RC-PDA-HG (Long et al., 2021) | 3664.0 | 7775.0 |
| | AdaBins (Bhat et al., 2021) | 3541.0 | 5885.0 |
| | P3Depth (Patil et al., 2022) | 3130.0 | 5838.0 |
| | LapDepth (Song et al., 2021) | 2544.0 | 5151.0 |
| | PnP (Wang et al., 2018) | 2496.0 | 5578.0 |
| | BTS (Lee et al., 2019) | 2467.0 | 5125.0 |
| | DORN (Lo & Vandewalle, 2021) | 2432.0 | 5304.0 |
| | RadarNet (Singh et al., 2023) | 2179.3 | 4898.7 |
| | CaFNet (Sun et al., 2024) | 2109.0 | 4765.0 |
| | Lin (Lin et al., 2020) | 1988.4 | 4841.1 |
| | SparseBeatsDense (Li et al., 2024b) | 1927.0 | 4609.6 |
| | RadarCam-Depth (Li et al., 2024a) | 1689.7 | 3948.0 |
| | GET-UP (Sun et al., 2025) | 1632.0 | 3932.0 |
| | TacoDepth (Wang et al., 2025) | 1492.4 | 3324.8 |
| | **POLAR (Ours)** | **1407.8** | **3193.5** |

Instead of relying on depth completion-style densification, POLAR directly learns a transformation function from radar to metric depth by leveraging polynomial fitting. Our method avoids the pitfalls of propagating unreliable local depth information by refining MDE predictions with learned polynomial coefficients, enabling flexible, scene-adaptive depth corrections that effectively capture object relationships and global scene structure.

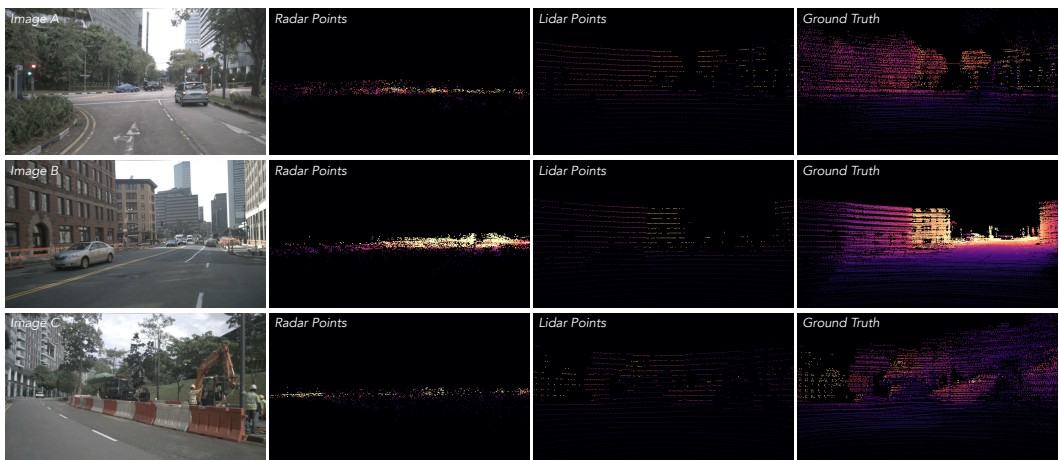

Figure 7: **nuScenes dataset visualization.** The elevation ambiguity of radar points results in erroneous projection onto the image plane that makes it challenging for depth completion methods to infer dense depth. In contrast, lidar points yield a denser, image-aligned projection, which is why accurate 3D scene reconstruction with depth completion methods is possible.

## F EVALUATION METRICS AND IMPLEMENTATION DETAILS

The evaluation metrics used in our study, Mean Absolute Error (MAE) and Root Mean Squared Error (RMSE), are formulated in Tab. 8. Lower values equal better performance for both MAE and RMSE. Unless specified otherwise, all reported values are in millimeters (mm). We train for 60 epochs using a cosine decay learning rate scheduler with learning rate of $5 \times 10^{-5}$, and use weighting terms $\lambda_{L_1} = 1.0, \lambda_{L_2} = 0.4, \lambda_{mono} = 0.25$ for our loss function.

## G ADDITIONAL COMPARISONS

**Full MDE Comparisons.** For ZJU (see Tab. 9), among all configurations, POLAR w/ UniDepth achieves the best performance, improving over raw UniDepth predictions by 61.1% and over RadarCam-Depth w/ UniDepth by 54.2%. For VoD (see Tab. 10), POLAR w/ UniDepth achieves the best performance in MAE, improving over raw UniDepth predictions by 42.4% and over RadarCam-Depth w/ UniDepth by 32.7%, while POLAR w/ Depth Anything achieves the best performance in RMSE, improving over RadarCam-Depth w/ Depth Anything by 37.6% and over raw Depth Anything predictions (inverted and median scaled) by 10.4%.

Table 9: **MDE backbone** comparative studies on ZJU-4DRadarCam (ZJU).

| Method | ZJU | |
| --- | --- | --- |
| | MAE | RMSE |
| DPT (Ranftl et al., 2021) | 1885.3 | 3326.1 |
| Depth Anything (Yang et al., 2024) | 1943.2 | 3469.3 |
| Depth Pro (Bochkovskii et al., 2024) | 1680.2 | 3144.9 |
| UniDepth (Piccinelli et al., 2025) | 1533.0 | 3188.4 |
| RadarCam-Depth w/ DPT | 1183.5 | 3229.0 |
| RadarCam-Depth w/ Depth Anything | 1724.4 | 3661.3 |
| RadarCam-Depth w/ Depth Pro | 1490.6 | 3429.5 |
| RadarCam-Depth w/ UniDepth | 1152.5 | 3168.6 |
| POLAR w/ DPT | 707.1 | 1216.9 |
| POLAR w/ Depth Anything | 657.2 | 1225.4 |
| POLAR w/ Depth Pro | 640.3 | 1174.8 |
| POLAR w/ UniDepth | **629.6** | **1171.3** |

Table 10: **MDE backbone** comparative studies on View-of-Delft.

| Method | View-of-Delft | |
| --- | --- | --- |
| | MAE | RMSE |
| DPT (Ranftl et al., 2021) | 4117.9 | 5498.9 |
| Depth Anything (Yang et al., 2024) | 3270.5 | 4411.9 |
| Depth Pro (Bochkovskii et al., 2024) | 3275.9 | 5936.7 |
| UniDepth (Piccinelli et al., 2025) | 2605.6 | 5691.0 |
| RadarCam-Depth w/ DPT | 4013.5 | 5911.9 |
| RadarCam-Depth w/ Depth Anything | 3103.6 | 6328.7 |
| RadarCam-Depth w/ Depth Pro | 2843.4 | 6082.0 |
| RadarCam-Depth w/ UniDepth | 2227.4 | 5385.8 |
| POLAR w/ DPT | 1891.4 | 4252.6 |
| POLAR w/ Depth Anything | 1770.5 | **3951.8** |
| POLAR w/ Depth Pro | 1520.2 | 3987.2 |
| POLAR w/ UniDepth | **1500.1** | 3960.5 |

**Leveraging Radar.** Tab. 13 shows that replacing radar points with learnable, dataset-specific points worsens MAE by 28.0% and RMSE by 29.9%, demonstrating that we indeed leverage the radar inputs effectively. As additional evidence, Tab. 11 shows that our method, evaluated zero-shot cross-dataset, achieves comparable or better performance than baselines trained on the target datasets. Tab. 12 shows we are more robust to reduced radar point density at inference, i.e., less performance degradation than the baseline method RadarCam-Depth.

Table 11: **Zero-shot** generalization.

| Method | nuScenes→ZJU MAE | RMSE | nuScenes→VoD MAE | RMSE |
|---|---|---|---|---|
| GET-UP (zero-shot) | 3845.2 | 8469.7 | 4809.1 | 8653.9 |
| RadarCam-Depth (zero-shot) | 5435.9 | 9785.8 | 7521.5 | 9194.8 |
| GET-UP (trained) | 1699.7 | 3882.6 | 2917.3 | 6145.1 |
| RadarCam-Depth (trained) | 1183.5 | 3229.0 | **2227.4** | 5385.8 |
| Ours (zero-shot) | **1147.9** | **3109.5** | 2256.2 | **4744.2** |

Table 12: Reduced radar point density.

| % radar kept / removed | RadarCam-Depth MAE | RMSE | Ours MAE | RMSE |
|---|---|---|---|---|
| 25% kept / 75% removed | 7969.4 | 10831.5 | 2416.4 | 4836.6 |
| 50% kept / 50% removed | 4819.8 | 7077.7 | 1816.8 | 3945.7 |
| 75% kept / 25% removed | 2537.3 | 4247.4 | 1575.2 | 3611.8 |
| 100% kept / 0% removed | 1689.7 | 3948.0 | **1407.8** | **3193.5** |

**Adapters.** Tab. 13 shows that recent adapter-based finetuning methods LoRA (Hu et al., 2022) and ViT-Adapter (Chen et al., 2022), even with a post-hoc linear fit to projected radar points, do not outperform us.

**Regression Baselines.** Isotonic regression and monotone spline fitting methods are natural baselines. Tab. 13 shows these methods for regressing projected radar points on MDE predictions do not outperform us. We hypothesize that this is due to noise in radar points that can be mitigated by learning (Sec. 3.2).

Our learned polynomial fit may, in principle, introduce unwanted inversions of initially correct MDE predictions. POLAR successfully mitigates this effect through the proposed novel first-derivative regularization term (see Sec. 3.4 and Eqs. 6, 7),

Table 13: Additional comparisons of POLAR vs. learnable dataset-specific points in place of radar points, and regression baselines.

| | MAE | RMSE |
|---|---|---|
| Learnable Points | 1860.9 | 4207.1 |
| Isotonic Reg. | 2895.2 | 4340.5 |
| Cubic Hermite | 2131.3 | 4588.0 |
| PCHIP | 1809.6 | 4054.7 |
| LoRA | 2030.6 | 4493.7 |
| ViT-Adapter | 1859.6 | 4280.0 |
| Our Performance | **1407.8** | **3193.5** |

which effectively constrains such inversions. To quantify this, we compute Kendall's $\tau$ coefficient between predicted and ground-truth depths. Our method achieves the highest monotonicity ($\tau = 0.969$) over regression baselines Isotonic Regression ($\tau = 0.871$), PCHIP ($\tau = 0.758$), and Cubic Hermite Spline ($\tau = 0.736$). The raw MDE predictions do exhibit monotonicity with respect to ground truth ($\tau = 0.957$), but our polynomial transformation increases it, indicating that we correct unwanted inversions. To assess statistical significance, we compute Kendall's $\tau$ over 30 bootstrap samples for both our method and the raw MDE predictions. A two-sample t-test reveals a statistically significant difference in mean monotonicity ($p = 0.012$).

# H PROOF: LIMITATIONS OF GLOBAL SCALE AND SHIFT

As further theoretical justification, we prove by construction that an affine scale-and-shift transformation is insufficient to fit MDE predictions to ground truth.

**Proposition 1.** *There exist infinitely many sets of $k \geq 3$ (MDE prediction $\hat{d}$, ground truth $d$) pairs such that no global scale $\alpha$ and shift $\beta$ satisfy $d = \alpha\hat{d} + \beta$ for all $k$ pairs simultaneously.*

*Proof by Construction.* Consider the following three (MDE prediction $\hat{d}$, ground truth $d$) pairs from the nuScenes dataset, specifically from the

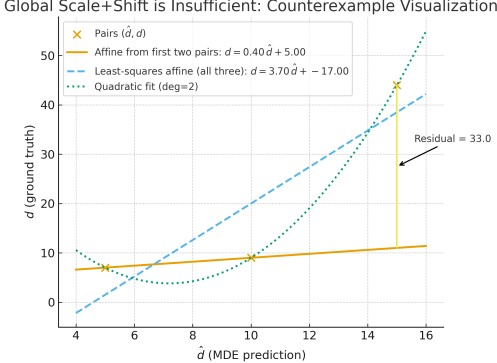

image shown in Fig. 3:

$$(\hat{d}, d) \in \{(5, 7), (10, 9), (15, 44)\}.$$

Assume to the contrary that there exist $\alpha, \beta \in \mathbb{R}$ such that

$$d = \alpha\hat{d} + \beta$$

holds for all pairs.

From the first two pairs, we obtain:

$$7 = 5\alpha + \beta, \quad 9 = 10\alpha + \beta.$$

Subtracting gives $\alpha = 0.4$ and $\beta = 5$ as the *unique* solution for these two pairs.

Applying this solution to the third pair yields:

$$\alpha \cdot 15 + \beta = 0.4 \cdot 15 + 5 = 11,$$

which contradicts the required equality with the ground truth value $d = 44$, since the residual error equals $44 - 11 = 33$ and not zero.

Hence no global scale $\alpha$ and shift $\beta$ exist that can satisfy all three pairs simultaneously. Moreover, scaling each pair by any nonzero constant produces infinitely many distinct 3-sets of $(\hat{d}, d)$ pairs for which no $\alpha$ and $\beta$ exists. Then, for any such 3-set, appending $k - 3$ arbitrary pairs yields an infinite family of $k$-sets ($k > 3$) that likewise admit no solution.

$\square$

**Corollary 1.** *It is therefore a misconception that the scale ambiguity in MDE can be resolved solely by a global scale and shift. In contrast, any smooth relationship between $\hat{d}$ and $d$ can be locally approximated by a polynomial via Taylor expansion, giving our polynomial fitting formulation the theoretical capacity to approximate $d$ as a function of $\hat{d}$ arbitrarily well.*

