# OpenReview forum: "Radar-Guided Polynomial Fitting for Metric Depth Estimation"
_ICLR.cc/2026/Conference — ICLR 2026 Conference Withdrawn Submission_

### Official Review · Reviewer_MTqC · 2025-10-18

**Soundness:** 3
**Presentation:** 3
**Contribution:** 3
**Rating:** 6
**Confidence:** 4

**Summary:**

This paper presents a RADAR-camera-fusion-based metric depth esitmation pipeline. Given a depth prediction from monocular depth estimation methods, the framework aims to re-scale the predicted depth to an accurate metric one. The major claim here is that the standard scale-and-shift alignment (i.e., projection) is not enough for an accurate output. The authors extend the scale-and-shift alignment (ax + b) to a polynomial one. And the network naturally is designed to predict the polynomial factors in a transformer style. A regularization term is added to make sure the polynomial function is monotonically increasing. Experimental results show good improvements.

**Strengths:**

- Good motivation. I'm not an expert in the area of this fusion-based depth esitmation field. But as far as I know, many depth estimation models predict affine-invariant depth and use scale-and-shift alignment for metric calculation. It would be good to point out that it scale-and-shift alignment is not enough for an accurate output.

- The overall framework is naturally and well designed with a clear motivation. Modules look seamlessly from my point view.

- Good experimental results. Improvement over previous methods is satisfactory.

- Well-written paper. It's easy for me to follow.

**Weaknesses:**

1. Ablation study

   Currently, all ablation study experiments are conducted based on a framework aiming to predict polynomial / scale and shift factors. It would be always good to see these results and the experimental results indicate the effectiveness of each module.

    However, all previous work fuse RADAR and camera in the feature space and directly predict the depth result, whereas the proposed framework aims to predict rescaling factors based on the monocular depth. It would be necessary to have an experiment to demonstrate a target switch like this can lead to performance gain. Though the authors provided something about Regression Baselines in Tab.13, it's not clear for me how this baseline is designed.

    I think it can be done by reversing the direction of the cross attention and adding one decoder for depth estimation.

    It would be better to put the ablation study in the main paper.

2. H PROOF is weird from my point of view. Scale and shift alignment is to estimate factors with a min global error and it's not for looking for best solutions without any error.

    The figure for H PROOF is super miss-leading, please remove the orange "affine from first two points" line. It's NOT how scale and shift alignment works at least.

    Adding reference for related work claiming alignment issue might also be helpful, like [1, 2]

    [1] benchdepth: are we on the right way to evaluate depth foundation models?
    [2] MoGe-2: Accurate Monocular Geometry with Metric Scale and Sharp Details

**Questions:**

Please check the weakness.

For me, it's hard for me to judge the novelty from an expert view in the field of fusion-based depth esitmation. I would like to check opinions from other reviewers who might have more experience.

It would be better to see a deeper discussion about the alignement issue in case we even don't have radar points as guidance (to somehow make broader impact for general depth field), or see some more rigorous experiments to demonstrate the point. If there are no other concerns from other reviewers and the authors can make this discussion better, I would like to increase the rating.

---

### Official Review · Reviewer_dVoq · 2025-10-30

**Soundness:** 3
**Presentation:** 3
**Contribution:** 2
**Rating:** 4
**Confidence:** 4

**Summary:**

This paper resolves scale ambiguity in monocular depth estimation by converting predictions to metric depth via automotive radar. It proposes POLAR, applying a learned polynomial transformation to pretrained MDE (Monocular Depth Estimation) outputs. Unlike global scale-and-shift (uniform adjustment), polynomial fitting introduces inflection points for non-uniform refinement. A lightweight multimodal network predicts coefficients from sparse radar and image features, with a novel monotonicity regularization (first-derivative penalty) to preserve local depth order.Evaluated on nuScenes, ZJU-4DRadarCam, View-of-Delft, POLAR achieves state-of-the-art accuracy: RMSE reduced by 30% on average (up to 50% on some datasets), with real-time inference (≈40 FPS) and lower computational cost than prior radar-camera fusion methods.

**Strengths:**

Polynomial fitting: Uses radar-guided high-degree polynomial transformation to align monocular depth to metric scale, enabling non-uniform adjustments to correct multi-object depth misalignments that affine transforms cannot fix.

Radar use: Leverages low-cost mmWave radar via lightweight feature fusion (no complex 3D fusion) to inform polynomial coefficients, gaining strong performance without heavy networks.

Efficiency: Streamlined single-stage design avoids heavy operations (e.g., explicit radar-image correspondence), achieving 24.8 ms/frame inference (40+ Hz) and reduced FLOPs for practical autonomous deployment.

**Weaknesses:**

Reliance on MDE local order: Assumes MDE provides reasonable local relative depth. Polynomial fitting (with monotonic constraints) cannot reorder local depths, so MDE’s local ordinal errors or structure misses remain uncorrected.

Dependency on radar quality: Radar’s sparsity/noise affects performance. No radar detections (range/reflectivity/occlusion) or outlier/noise reduce polynomial guidance, degrading scaling accuracy in affected depth ranges.

**Questions:**

Handling local depth errors: How does POLAR perform if MDE has local mistakes (e.g., object predicted closer than background)? Does monotonic constraint block error correction? Are there observed failure modes from this assumption, and how to address them (e.g., adding local depth inversion detection) ?

---

### Official Review · Reviewer_F7hX · 2025-11-01

**Soundness:** 2
**Presentation:** 3
**Contribution:** 3
**Rating:** 4
**Confidence:** 3

**Summary:**

In this paper, the authors propose a method for depth estimation using sparse radar points. They use an off-the-shelf scale-less depth estimator, then correct its depth predictions with polynomial coefficients predicted from both input radar points and their relationship with the initial depth estimation. The authors demonstrate good performance against a number of prior works, ablate their key choices, and do due diligence to report against depth completion focused works as well.

**Strengths:**

- This paper proposes an interesting framework for polynomial-based correction of scale-less depth predictions from an off the shelf network.
- The authors apply this method to radar input points, demonstrating effective depth correction and improvement with more coefficients.
- This reviewer found the monotonic regularization a good insight.

**Weaknesses:**

- The main results in Table 1 appear to use UniDepth (Table 4). However, I think the TacoDepth results use DPT (explained in Table 1 of TacoDepth). Using DPT instead, the authors report 1525.6/3745.0 MAE/RMSE, which is worse than TacoDepth's 1492.4/3324.8. This raises a bit of a concern of fair evaluation. A similar concern also affects ZJU.
- Why did the authors decide to develop such a specific type of feature aggregation method in Equation 2? For instance, given the similarities between L2 distance and dot product (similar w/ normalization), this could could have been a standard cross-attention layer instead. This ablation could be useful.
- Equation 3 seems like a standard cross-attention layer, moving features from prototypes to the image. Why are the prototypes necessary at all? Would not a cross attention layer directly from radar points to the image suffice, when adjusted for the same # of parameters used? An ablation could give more insight.
- While the polynomial approach is useful for aligning the overall scene depth, it is, to the best of my knowledge, still a region-agnostic correction method. To this reviewer, it seems like instead of having the same correction for all scene elements of the same predicted depth, instead, different scene elements should have their own correction.
- For the results in Table 4, is the method trained independently for each depth estimator, or is a single model used for all? If the former, how might the latter work, to evaluate generalizability of the model to different depth estimators?
- A core argument in the paper seems to be that such polynomial fitting/correction is better than direct depth prediction. I believe an ablation demonstrating this, possibly by trying to make direct depth prediction in the 2D image plane after the radar-to-image cross attention, would strengthen the paper.
- While the authors did due diligence to compare with depth completion methods, there are a number of works focusing on sparse or unevenly distributed depth input that should probably be referenced [1-7]. While not radar, and it's likely too much to request comparison with these works during the rebuttal phase, some discussion of the ideas in these papers and how they may or may not be applicable could strengthen this work.
- Building on the previous point, in Table 7, did the authors re-train the other methods? I find it surprising, for instance, that NLSPN which uses depth information fares worse than BTS, which is a pure monocular model.

[1] Towards 3D Scene Reconstruction from Locally Scale-Aligned Monocular Video Depth (https://arxiv.org/pdf/2202.01470v3)
[2] Sparsity Agnostic Depth Completion (https://openaccess.thecvf.com/content/WACV2023/papers/Conti_Sparsity_Agnostic_Depth_Completion_WACV_2023_paper.pdf)
[3] Sparse SPN: Depth Completion from Sparse Keypoints (https://arxiv.org/pdf/2212.00987)
[4] Flexible Depth Completion for Sparse and Varying Point Densities (https://openaccess.thecvf.com/content/CVPR2024/papers/Park_Flexible_Depth_Completion_for_Sparse_and_Varying_Point_Densities_CVPR_2024_paper.pdf)
[5] SparseDC: Depth Completion from sparse and non-uniform inputs (https://arxiv.org/pdf/2312.00097)
[6] SparseFormer: Attention-based Depth Completion Network (https://arxiv.org/pdf/2206.04557)
[7] Towards Domain-agnostic Depth Completion (https://arxiv.org/pdf/2207.14466)

**Questions:**

- [Minor] L235: Just to confirm, "projected radar features" refers to radar features "projected" using the MLP, not that the radar points themselves are projected to 2D?
- I would appreciate it if the authors could address my concerns above; for instance, some additional ablations could be helpful. Some core ideas, such as the polynomial fitting, remain not sufficiently tested, and I have some concerns about comparison fairness with TacoDepth. Given these concerns, I give an initial rating of 4, but I am open to raising my ratings if the above concerns are adequetely addressed.

---

### Official Review · Reviewer_3ts8 · 2025-11-02

**Soundness:** 2
**Presentation:** 3
**Contribution:** 3
**Rating:** 4
**Confidence:** 3

**Summary:**

The paper tackles the metric depth estimation by leveraging the radar input and a pre-trained universal monocular depth estimator (MDE). The proposed POLAR, which starts from a frozen MDE and predicts a scene-specific polynomial. Coefficients are predicted from fused radar–image features; a first-derivative regularizer encourages the mapping to be “approximately monotone.” The method reports competitive accuracy and efficiency on several benchmarks, including nuScenes, ZJU-4DRadarCam, with best results around degree-8 polynomials.

**Strengths:**

1.  The idea of performing the scale correction as a non-linear, scene-level polynomial fit to an MDE’s output is clean and practically attractive. It departs from the dominant “decode dense depth from fusion” paradigm.

2. The proposed method has strong empirical results. For instance, it shows consistent gains across multiple datasets and methods (e.g., Depth Pro, UniDepth). Ablations on polynomial degree and components are informative

3. The paper is well-written and easy to follow.

**Weaknesses:**

1. The mapping is global 1D, which is kind of limited.  Using a single scene-global polynomial maps equal (z) values to the same D everywhere. It cannot fix spatially entangled errors where two regions share similar predicted (z) but need different corrections (e.g., reflective car vs. dark wall).

2. Inefficient ablation studies. Beyond showing that affine mappings can fail, there is insufficient analysis regarding the identifiability. eg, under radar sparsity/noise, stability vs. polynomial degree.

3. The efficiency table is ambiguous. It’s unclear whether latency/GFLOPs include the MDE backbone for all methods. Provide backbone cost, fusion/decoder cost, and true end-to-end numbers with GPU/precision/batch/resolution details.

**Questions:**

see above weakness

---

### Note · Authors · 2025-11-14

I have read and agree with the venue's withdrawal policy on behalf of myself and my co-authors.